computer modelling and simulation/complexity

social care, kinship networks, agent-based modelling

**Author for correspondence:**
Eric Silverman
e-mail: eric.silverman@glasgow.ac.uk

# Modelling social care provision in an agent-based framework with kinship networks

## Umberto Gostoli and Eric Silverman

MRC/CSO Social and Public Health Sciences Unit, University of Glasgow, Glasgow G2 3AX, UK

ES, 0000-0003-0147-6118

Current demographic trends in the UK include a fast-growing elderly population and dropping birth rates, and demand for social care among the aged is rising. The UK depends on informal social care—family members or friends providing care—for some 50% of care provision. However, lower birth rates and a greying population mean that care availability is becoming a significant problem, causing concern among policy-makers that substantial public investment in formal care will be required in decades to come. In this paper, we present an agent-based simulation of care provision in the UK, in which individual agents can decide to provide informal care, or pay for private care, for their loved ones. Agents base these decisions on factors including their own health, employment status, financial resources, relationship to the individual in need and geographical location. Results demonstrate that the model can produce similar patterns of care need and availability as are observed in the real world, despite the model containing minimal empirical data. We propose that our model better captures the complexities of social care provision than other methods, due to the socioeconomic details present and the use of kinship networks to distribute care among family members.

## 1. Introduction

In recent history, researchers and practitioners in public health have succeeded in significantly lengthening the human lifespan and increasing quality of life for the elderly. However, as human lifespans continue to lengthen and birth rates drop throughout much of the developed world, many nations are experiencing an increase in demand for social care—the provision of personal and medical care for people in need of assistance due to age, disability or other factors. In the UK, the elderly consume the largest share of social care, and lower birth rates mean that the supply of available carers is decreasing over time even as demand is

growing rapidly [1]. As a result, social care is a frequent topic in UK policy debates, with widespread concern that the country will be unable to afford the significant public investment needed to address this growing care need.

Age UK notes that nearly 50% of those aged 75 and over are living with a long-term limiting health condition—and the fastest-growing age group in the UK is the over-85s, so the problem will only get worse in the coming decades [2]. According to the Health Survey for England 2017, 14% of adults aged 65–69 need help with one or more activities of daily living (ADLs) or instrumental activities for daily living (IADLs), a figure rising to 44% for adults aged 80 and over [3]. As a result, social care need is rising at a pace that outstrips the growth of public and private care supply. Unmet social care need is therefore an increasingly important and widespread social problem in the UK. The 2017 Ipsos MORI report *Unmet Need for Care* showed that 'more than half of those with care needs had an unmet need for at least some of their needs' [4]. This means that less than half of those elderly individuals in need of assistance with ADLs were able to receive sufficient care. Large numbers of UK citizens are thus living without sufficient care; an estimated 1.2 million people did not receive sufficient care for their needs in 2017, an increase of 48% since 2010 [2].

The increased pressure on the social care system also has an impact on the healthcare system, with delayed discharges from hospital being a particularly expensive consequence of unmet care need. Patients in need of social care frequently stay in the hospital longer than necessary due to a lack of care availability, leading to a shortage of beds for other patients and increased costs to hospitals. According to Age UK, between 2010 and 2016, the number of additional bed-days attributable to a lack of available home-care packages increased by 181.7%, while wait times for residential care placements increased by 40% [2]. In 2016, the National Audit Office estimated the total delays attributable to care shortages amounted to 2.7 million bed-days per year, resulting in an annual cost to the National Health Service (NHS) of approximately £820 million [5].

The UK is largely dependent on informal social care, or care provided free-of-charge by family members and loved ones, in order to meet the needs of the population. Informal care is enormously widespread in the UK and is much larger than the formal care infrastructure. A 2018 report of the National Audit Office estimates the value of informal care in the UK as being as high as £100 billion per year, about 10 times the total value of the care arranged by local authorities, estimated to be around £20.4 billion in the period 2016–2017 [6]. The Family Resources Survey 2013/2014 showed that there were 5.3 million informal carers in the UK [7], while projections indicate that the number of people receiving informal care will increase by 60% in the period 2015–2035 [8]. According to the Health Survey for England 2017, 68% of participants aged 65 and over reported receiving help from unpaid helpers, while 21% said they had received help from both unpaid helpers and paid helpers [3].

Looking beyond informal care, Health Survey for England 2017 findings confirm that paid-for (or formal) care has an important but minor role, as only 16% of adults aged 65 and over reported receiving privately paid-for care and just 5% reported receiving help from their local authority [3]. The National Audit Office estimates the total privately paid-for care in the period 2016–2017 at approximately £11 billion. This figure increases to approximately £14 billion when private 'top-ups' are considered; these are additional private supplements to care provided by local authorities, which totals approximately £15 billion in the same period [6].

Given current UK demographic trends, the increase in demand is such that Carers UK proposes that carer numbers would need to increase 40% over the next two decades to meet demand [9]. At the same time, according to Wittenberg & Hu [8], the number of people using privately funded social care is expected to rise by almost 50% by 2035, and private expenditure for social care is projected to increase from £6.8 billion to almost £20 billion in the same period, almost a three-fold increase.[1]

Understanding how the need for social care evolves and the process through which informal and formal care is provided is a vital component in developing sustainable social care policies. In this regard, the importance of support and care-giving networks has long been recognized [10,11]. In particular, research has shown that informal care is provided mostly through care networks with an average of three to five members [10]. These networks are predominantly composed of an individual's close relatives. Aldridge & Huges [12] report that 72% of carers provide care to a member of their immediate family, whether a parent (40%), partner (18%), son or daughter (14%), while Petrie & Kirkup [13] show that 51% of carers

---

[1]We note that these projections are based on certain assumptions: 'The base case projections take account of expected changes in factors exogenous to long-term care policy, such as demographic trends. They hold constant factors endogenous to long-term care policy, such as patterns of care and the funding system' [8] (p. 4). We thank an anonymous reviewer for pointing out the specific base case assumptions on which these projections are based.

provide care to someone in their household. Wettstein & Zulkarnain [14] show that, in 2011, 31% of informal care in the US was provided by spouses, 47% by children and 18% by other relatives (sons-in-law, daughters-in-law and grandchildren), with only 4% being provided by non-relatives.

In addition, empirical research has shown that the kind of social care provided is affected by socioeconomic status. Petrie & Kirkup [13] report that people working in routine occupations and those with lower qualifications are more likely to provide informal care. Moreover, Laing & Buisson [15] find that, while across the UK, an estimated 24% of care home residents are funded through private 'top-ups', in North-East England, only 18% are funded privately, as opposed to 54% in the wealthier South-East. Overall, these findings suggest that informal care becomes less common in higher socioeconomic status groups than in lower groups, while formal care becomes more common.

Finally, the social care literature also reveals a significant gender gap in social care provision. Wettstein *et al.* [14] reported that in the US, daughters are almost twice as likely (31%) to provide care as compared to sons (16%). These findings are confirmed by Petrie & Kirkup [13], who find that 59% of family carers in the UK are women. At the population level, 16% of women provide informal care as compared to 12% of men.

In this paper, we propose an agent-based model of the UK informal and privately funded formal social care system. This model reflects the complexity of a system where demographic, social and economic processes interact to determine the dynamics of social care demand and supply. Our aim is to provide a theoretical framework that allows us to improve our understanding of the mechanisms driving unmet social care need. Using ABMs enables us to model scenarios of economic and social policy change in virtual populations, providing a means to test complex policies targeted at multiple levels of society. Such models can allow policy-makers to experiment with potential policy interventions and reveal any possible unintended side-effects of those policies prior to implementing them in the real world.

Previous work has attempted to address the social care problem using agent-based simulation approaches [16,17]. Here we present a simulation that significantly extends these previous efforts. The model provides a more comprehensive simulation of social care provision behaviour, via the inclusion of a detailed socioeconomic model and the representation of care provision as not just a simple one-to-one exchange of resources, but a complex negotiation taking place between members of the care receivers' kinship networks.

# 2. The model

## 2.1. Motivations

Here we present an agent-based model as a formal representation of the complex demographic and social processes affecting informal social care demand and supply, and the dynamics of the social care outcomes resulting from their interaction over time. Our primary concern at this stage was not the precise replication of empirical, real-world data but the development of a theoretical framework potentially capable of representing the full complexity of the social care system. In our results, we aim for *qualitative similarity* to real-world social care trends, rather than precise numerical replication, in order to determine if our modelling of the underlying processes is producing appropriate outcomes.

With these motivations in mind, at this initial stage, the behavioural assumptions made in this paper should not be considered necessarily accurate, but rather should be seen as a first approximation, i.e. a necessary point of departure allowing us to provide proof-of-concept results and demonstrate the model's potential as a policy-making tool. Further work and more specialized expertise will be needed to refine and revise the model's behavioural assumptions in order for the model to be used in this way. In particular, in this model, we do not consider the role of individual attitudes and preferences in the choice of types of care sought and provided, though these elements can be included in future iterations. In the current formulation, agents make choices about care based on their socioeconomic status and their households' demographic structure.

Ultimately, our aim is for this simulation to facilitate the development and evaluation of alternative social care policies, including complex interventions aimed at behavioural change. Our addition of healthcare costs and a detailed economic model to the simulation can also clarify the impact of interventions on other key areas of policy, enable more robust policy evaluation and reduce the incidence of unintended consequences derived from otherwise well-meaning policy prescriptions.

Future versions of this framework can also be applied to nations other than the UK by altering the structure of the modelled care system and the simulated geography and population data.

## 2.2. Basics of the model

The model itself is complex and contains many detailed economic and social processes and sub-processes; this section provides a high-level overview of the model's functionality. For details of the previous models which formed the initial basis for this simulation, please refer to Noble et al. [16] and Silverman et al. [17].[2]

The simulated agents occupy a space roughly based on UK geography. Agents live in houses which form towns containing clusters of up to 1225 houses. These clusters vary in size in rough proportion to real-world population density, which varies across the 8 × 12 grid composing the model's geography. The agent population is scaled down from real UK levels at a factor of roughly 1:10 000.

The model updates in 1-year time steps. Initial populations are generated and randomly distributed in the year 1860, and the model then runs until 2040 when final social care costs are calculated and recorded. We start the model in 1860 to ensure the population dynamics have time to settle before empirical population data (UK Census data) are integrated into the model in 1951.

## 2.3. Agent life-course

Newborn agents are classed as dependent children until age 16, at which point they reach adulthood (the minimum working age in the UK). The agents then decide whether to continue to study or to start looking for a job. This choice is repeated every 2 years until the age of 24. This is a probabilistic choice that depends on the household's income level and on the parent's education level. When the agent decides to look for a job (or when they reach age 24 at the latest), the agent then enters the workforce. During their working life, agents can be hired, fired and can change jobs. When an agent is unemployed, they find a job with a certain probability which depends on the unemployment rate (which is an input of the model), then start earning an income and paying tax. At a certain age, agents retire from the workforce (at age 65 when default parameter settings are used), at which point they stop paying tax and begin receiving a pension.

This version of the simulation uses a Gompertz–Makeham mortality model to approximate mortality rates until 1951, as in Noble et al. [16]. From 1951, the simulation uses mortality rates from the Human Mortality Database [19]. After 2009 a Lee–Carter model is used to generate future mortality rates, as in Silverman et al. [17].

### 2.3.1. Partnership formation

Upon reaching adulthood, agents can form partnerships with other agents.[3] Employed male agents and adult female agents are randomly paired with probabilities based on the agents' socioeconomic difference, age difference and geographical distance (with the relative weight of these three factors depending on the model's parameters). Age-specific annual divorce probabilities determine whether a couple dissolves their partnership. Fertility rates follow the procedures outlined in Silverman et al. [17], in which data from [20] and the Office for National Statistics [21] are used from 1950 until 2009, at which point Lee–Carter projections are used.

### 2.3.2. Migration

As in the real world, agents can migrate for a variety of reasons. An agent becomes independent when they leave the parental home. This can happen when they form a partnership, when they find a job in a different town from the family home, or when they find a job in the same town of their parental home. When a partnership dissolves, the male agent will move elsewhere on the map, while any dependent children resulting from that partnership will stay with the mother. A family will relocate to a new house if one of the two parents finds a job in another town or when the family needs a larger house due to the increased size of the family. Retired agents with social care needs may elect to move into

---

[2]For those who wish to examine the model more closely, or run it for themselves, you can find the annotated Python 2.7 code available in [18].

[3]Hereafter we use 'partnership' as shorthand for relationships capable of producing children.

**Table 1.** Care need categories, with the number of hours of care required per week.

| care need category | weekly hours of care required |
| --- | --- |
| none | 0 |
| now | 8 |
| moderate | 16 |
| substantial | 32 |
| critical | 80 |

the household of one of their children, with a probability directly proportional to the amount of care provided by that household. Rarely, dependent children will be orphaned before reaching adulthood; in that case, the agent will be adopted by a household of their kinship network, or by a randomly selected couple if no such household is available.

## 2.4. Health status and care need

Agents begin in a healthy state and have no need of additional care. They may enter different categories of care need depending on age- and gender-specific probabilities.[4] Table 1 shows the five possible categories of care need and the amount of care required per week at each level of need. Agents who enter a state of care need do not recover and return to normal health but instead continue to progress to higher levels of need over time. The probability of progressing to higher care need levels depends positively on the agent's age and on the discounted sum of the agent's unmet care need in past periods.[5] In this model, the agents' level of unmet care need affects also their death probability (together with their age, SES and care need level).

Social care provision is linked to informal care availability in an agent's kinship network. An agent's kinship network's nodes consist of the households of agents with a familial relationship with the agent; the degree of kinship is defined as the network distance between the household and the agent. If they have time or available income, agents will provide informal or formal care to anyone in their kinship network with care need. The amount of care agents are available to provide depends on their status (through their income), their kinship relationship with the receiver and their geographical distance from the receiver (see the Kinship Network subsection below for details).

## 2.5. Model enhancements

This updated version of the Linked Lives model presented in Silverman *et al.* [17] has been rewritten from the ground up and substantially extended for greater detail and realism. The following features have been introduced:

— The population is composed of five *socioeconomic status groups*.
— Care supply is provided by an agent's *kinship network*, a network of households which have a kinship relationship to the agent.
— Households allocate part of their income to care provision, which can be in the form of both informal and *formal* care.
— *Government-funded social care* is introduced, whose structure reflects that of public social care in the UK (with the exception of Scotland).
— A *salary function* implying an inverse relationship between the time taken off work to provide informal care and the agent's hourly wage.
— Unmet social care needs affect the agents' *hospitalization* probability and the associated healthcare costs.

[4]Note that in the UK, people with critical care needs are likely to be placed into care homes. In this model, we do not explicitly represent this aspect, but, for the sake of simplicity, the social care received in a care home is implicitly considered as part of the formal social care received by people in need. Future iterations of the model will simulate care homes more explicitly.

[5]Our assumption, informed by consultation with social care experts, is that a prolonged period of unmet care need will increase the agents' frailty and therefore the probability that their health will further deteriorate.

## 2.5.1. Socioeconomic status groups

The population is composed of five distinct socioeconomic status (SES) groups. These categories follow the Approximated Social Grade, a socioeconomic classification produced by the Office for National Statistics, which is composed of six categories (A, B, C1, C2, D and E). For convenience, we redistributed category E (state pensioners, casual and lowest grade workers, unemployed with state benefits only) into the categories D (semi-skilled and unskilled manual workers) and C2 (skilled manual workers), to maintain a unimodal distribution. We initialized the groups' distribution to roughly reflect the 2016 UK distribution. The SES groups are characterized by different education levels, wages, career paths (represented by income growth curves) and unemployment rates. Moreover, the agents' SES affects their wealth, which is assigned to agents according to their accumulated income in order to replicate the 2017 UK individual wealth distribution.

The introduction of SES groups has a number of effects on the various stages of agent life courses. A higher socioeconomic position is associated with lower mortality and fertility rates and to a lower probability of developing care need. In the marriage market, the probability that two opposite-sex individuals will form a couple depends on their SES distance (which determine the marriage probability together with the partners' geographical distance and age difference).[6] In the job market, a higher SES is associated with higher starting and maximum salaries (but a lower salary growth rate); a higher probability to find a job and a lower probability to be fired (probabilities which are reflected in a lower unemployment rate) and a higher probability to change jobs if the job offer comes from a different town from their current hometown.

Given that in this model care supply and relocation decisions depend on income level, the socioeconomic position of an agent affects its behaviour (and that of the household it belongs to) through the agent's income, as a higher socioeconomic position is associated with a higher income level. For example, the share of income the household allocates to care supply depends positively on the household's *per capita* income (see the Formal Care section).

We included an inter-generational mobility process which allows agents to move to a different SES group from their parents'. Each SES group is associated with an education level. From the age of 16, an agent can decide whether to continue its studies (a choice that will allow the agent to reach a higher education level and therefore a higher SES group) or start searching for a job (in which case the agent is assigned the SES group associated with the education level reached). This choice is made by the agents every 2 years, until the age of 24 (i.e. at ages 16, 18, 20 and 22).[7] The probability that an agent keeps studying depends on three factors: the *per capita* available income (i.e. net of social care costs) of the agent's household; the difference between the maximum education level reached by the agent's parents and the agent's current education level and the amount of time the agent allocates to informal social care supply. In this highly stylized inter-generational mobility process, an agent's SES group is determined by the education level the agent reaches.[8]

## 2.5.2. Kinship networks

Each agent is associated with a kinship network—a network of households containing at least one agent with a kinship relation to them. This network includes:

— the agent's household (distance 0);
— the households of the agents' parents and of children who are not part of the agent's household (distance I);
— the households of the agents' grandparents, grandchildren, brothers and sisters who are not part of the agent's household nor of the households at distance I (distance II);
— the households of the agents' uncles, aunts, nephews and nieces who are not part of the agent's household nor of the households at distance I and II (distance III).

The kinship network has two functions. First, the network defines the total care supply available for a particular agent in need. The care receiver's total care supply is the sum of the available care supply

---

[6]The relationship between SES distance and probability of forming a couple is asymmetric: the probability of getting married decreases less rapidly with the SES distance if the higher-status individual is male rather than female.

[7]We assume each education step lasts 2 years. The educational stages correspond to A-level, Higher National Diploma, Degree and Higher Degree.

[8]Meaning the SES group of agents ending their studies when they are 16, 18, 20, 22 and 24-years-old, will be, respectively, D, C2, C1, B and A.

**Table 2.** Amount of care agents can provide depending on their status and distance from the receiver.

| agent status | household (D-0) | D-I | D-II | D-III |
| --- | --- | --- | --- | --- |
| teenager | 16 | 0 | 0 | 0 |
| student | 16 | 8 | 4 | 0 |
| employed | 16* | 8* | 4* | 0* |
| unemployed | 28 | 16 | 8 | 4 |
| retired | 56 | 32 | 16 | 8 |

*These are the minimum amounts the employed can offer, representing the informal care provided outside of working hours. If needed, additional informal care can be supplied by these agents taking time off their working hours. They can also use their income to pay for formal care. See the Formal Care section for details.

of all the members of all the households that are part of the kinship network which met certain conditions. These conditions represent kinship- and space-specific limitations in the supply of care. Kinship relationship and distance determine the quantity of informal care that each member of the receiver's kinship network is available to supply, as shown in table 2:

In particular, we assume first that the amount of care which an agent is available to provide to another agent depends positively on the closeness of their kinship. Agents living with the care receiver are an exception, as these agents' available care supply is assumed to be equal to that of the receiver's next of kin (i.e. spouse, children and parents), independent from the kinship degree (although most of the time the members of the receiver's household are his/her next of kin). The second factor affecting the informal care supply availability is the physical distance from the receiver's household. With regard to the physical distance, we can distinguish three classes of care suppliers:

— agents living in the receiver's household;
— relatives living in another household in the same town as the care receiver;
— relatives living in a different town.

We assume that only households living in the same town of the care receiver can provide informal care. In the case of formal care, we assume that only the care receiver's household and the households at distance I (i.e. the care receiver's parents or children) are available to provide formal care.

The care allocation process proceeds in a series of iterations in which a 4-h 'quantum' of care supply is transferred to an agent needing care from one of the households with available supply in its kinship network. The care allocation function first samples a care receiver from the pool of people with unmet social care need who have a kinship network with available care supply. The probability of each care receiver being sampled is directly proportional to the care receivers' quantity of unmet care need. Then the allocation function samples a household from the selected care receiver's kinship network with a probability that is directly proportional to the household's available supply. Once the supplying household has been selected, a 'quantum' of care is transferred from one of the household's members with available supply to the care receiver.

The member who is to provide care within the selected supplying household is determined in two steps. First, one of the household's six possible care sources is selected with a probability proportional to the residual available care of each source. The six care sources consist of the five groups that can provide informal care as shown in table 2, plus a sixth source which represents the amount of care that can be supplied by allocating part of the household's income (a category we call *out-of-income care*). The household's out-of-income care supply is the share of income that the household has available to allocate to care, either directly in the form of formal care, or indirectly in the form of informal care provided by employed members taking unpaid time off work. If one of the first five sources is selected, the household member with the greater residual available care within that category will provide care. If the out-of-income category is selected, the household will provide formal care if the lowest hourly wage among the hourly wages of the employed household members is higher than the hourly cost of social care. Otherwise, the employed household member with the lowest hourly wage will provide the 'quantum' of care.[9]

[9]Given that the quantity of out-of-income care supply depends on the household's *per capita* income, this care allocation mechanism implies that the probability that members of the households will spend time providing informal care is inversely related to the household's *per capita* income. In other words, the wealthier the supplying household, the more likely that it will provide formal care.

At the end of this step, both the care receiver's care need and the supplying household's availability are reduced by the amount of care transferred (the 4-h 'quantum' of care). The allocation process is repeated until the set of care receivers with unmet care need *and* available care supply is empty (i.e. there are no more care receivers with outstanding care need and available care supply). The ultimate result of this care allocation function is that units of social care are distributed from potential providers to receivers across the receiving agents' kinship network, and decisions about who provides care and choosing informal or formal care provision are made according to the composition, socioeconomic position and employment status of potential care providers. We suggest this detailed decision-making process better represents the complexities of care decisions that families need to navigate. The specifics of this complex process can be adjusted further by incorporating insights received from qualitative and quantitative data on informal care-givers and their decision-making.

The kinship network also allows households to compute the *informal care attraction* associated with each town, which represents a rough measure of the informal care that a household expects to get or supply in a given town. The households use this town-specific attraction to make relocation decisions.[10] More precisely, for a particular household $H_i$, the informal care attraction of a town $T$ is a function of the sum of the members of the households with a kinship relationship with $H$ living in the town $T$.[11] The contribution of a household $H_j$ to the social care attraction is weighted according to the degree of kinship between $H_i$ and $H_j$. This weighted sum is then multiplied by the complement to one of the shares of care provided by the government. Two assumptions characterize the towns' informal care attraction: first, the larger the local kinship network in a town (in terms of number of people who are part of it), the higher the amount of social care the agent can expect to receive (or to supply) and the higher the social care 'value' associated with a town; second, the higher the share of care supplied by the government, the lower the importance of the potential informal care in the relocation decision.

Apart from this 'network size' factor, a household's probability of relocating depends on the relocation cost, which increases with the household's size and the number of years the household's members lived in the current town[12], and the town's *homophily* attraction, which depends on the town's share of people belonging to other SES groups. Towns with a larger share of unoccupied houses also are more likely to be chosen for relocation.

### 2.5.3. Formal care

Both informal and formal care are allocated through the care recipient's kinship network. Each household allocates a share of its income to care, which increases with the household's *per capita* income.[13] We assume that formal care can be provided only by the care receiver's own household or from the households in the care receiver's kinship network with distance equal to 1.[14] As explained above, the choice between the informal and formal care is stochastic, with probabilities equal to the relative availability of available time and income allocated to social care, respectively. In order for the household to supply formal care, the hourly wage of the working member with the lowest wage (and available time left) must be higher than the price of formal social care. If the hourly wage of this member is lower than the price of formal social care, they will take time off work to provide informal care, as in this case, it is cheaper for the household to give up the agent's salary than pay for formal care. However, the supplying household can provide informal care only if it is in the same town of the care receiver, otherwise it can only provide formal care.

---

[10]In this model, agents face the choice of relocating to other towns if they receive a job offer or partner with someone from another town.

[11]The household's kinship network is obtained by combining the kinship networks of the household's members.

[12]The relocation cost $R$ can be thought of as a measure of the social capital developed by the household in their current town. This social capital would be lost by relocating to another town, so it acts as a barrier to relocation. Formally, relocation cost is computed as:

$$R = K \sum_{i=1}^{n} y_i^p,$$

where $n$ is the set of the household's members, $y_i$ is the number of years the household member $i$ spent in the current town and $p$ is a parameter with a value smaller than 1 (as additional years will increase the member's social capital by increasingly smaller amounts). $K$ is a scaling parameter.

[13]More precisely, with $x$ being the household's *per capita* income, the share allocated to care supply is the complement to one of the reciprocal of the exponential function of the product of $x$ and a parameter (the *incomeCareParam* included in the sensitivity analysis presented in the Results section).

[14]In other words, only the care receiver's parents and children are available to pay for formal care if they cannot provide informal care.

### 2.5.4. Government-funded social care

We introduce a stylized government-funded social care scheme which mirrors the public social care scheme in force in the UK, apart from Scotland.[15] In line with this policy, all adults with a critical level of care need and whose level of savings is below £23 250 receive some public financial support.[16] If their savings are below £14 250, the government pays all the social care expenses the care receiver cannot pay without reducing his income below £189 (called the *minimum income guarantee*), whereas above this level of savings, the amount paid by the government is reduced by £1 for every £250 of savings.

### 2.5.5. Salary function

We model the hourly salary that an agent receives (on average) when it is employed, using the Gompertz function. This function is a double exponential that takes the following three SES-specific arguments:

— the initial salary level;
— the final salary level;
— the salary growth rate.

The salary growth rate is multiplied by the agent's cumulative work experience, which is the discounted sum of all the fractions of the working week allocated to work (if an agent works full time, this fraction is equal to 1).[17]

This equation implies that if an agent takes time off work to provide informal care, this will result in less work experience and therefore a lower hourly salary. Given the properties of the care allocation mechanism, the agents employed with the lowest hourly salary will be more likely to provide informal care in the future (if there are not retired/students/unemployed agents or if the household's retired/students/unemployed agents do not have available informal care supply). In this model, new mothers devote their time to caring for their newborn, meaning that we see a gender pay gap emerge due to the interaction between:

— the allocation of part of the household's income to care supply;
— the choice of the caregiver within the supplying household;
— the salary function.

Retired agents receive a pension which is proportional to their final income level. We assume that when an agent needs care they retire due to sickness and thereafter receive a pension. If their care need is low or moderate, their ill-health pension is their normal pension reduced in accordance with the ratio between lost working years and the maximum number of working years (in other words, the working years of a person in good health). If the agent's care need is substantial or critical, the ratio is computed by reducing the number of lost working years by 50%.

### 2.5.6. Hospitalization

We assume that agents with care need will spend additional time in the hospital; the duration of their stay is a function of the agents' care need level and the average discounted share of unmet care need. The higher the agent's care need level and the higher her average share of unmet care need, the greater the number of days the agent is expected to spend in the hospital each year. By multiplying the sum of the hospitalization duration by the hospitalization cost per day, we can determine the cost of unmet care need for the National Health Service.

---

[15]For the sake of simplicity, at this stage, we will not differentiate policies by region, although the spatially explicit framework we adopt makes this future development quite straightforward.

[16]In this model, public financial support is used for generic public formal care without distinguishing the care received at home from that received in care homes.

[17]Formally, the salary function is:

$$w = Fe^{ce^{-rt}},$$

where:

$$c = \ln\frac{I}{F},$$

and $F$ is the maximum hourly wage, $I$ is the initial hourly wage, $r$ is the wage growth rate and $t$ is the discounted cumulative work experience.

### 2.5.7. Simulation steps

Each 1-year time step of the simulation unfolds through the following sequence of 11 steps:

(1) *Deaths*: agents are removed from the population with a given probability depending on age, SES, care need level and unmet care need;
(2) *Adoptions*: children without parents are adopted;
(3) *Births*: married women give birth to new agents with a probability depending on age and SES;
(4) *Divorces*: some couples dissolve their partnership and the male agent relocates;
(5) *Marriages*: some singles get married and go to live together with a probability depending on age, SES and geographical location;
(6) *Social Care Allocation*: social care is transferred from agents with available hours (or income) to agents with social care need;
(7) *Age Transitions*: the age of the agents is incremented, and their age-related status is updated;
(8) *Social Transitions*: students decide whether to start working or keep studying, depending on their family income per capita, their parents' SES and their care responsibilities; if they start looking for a job, they are assigned the SES associated with the education level they have reached;
(9) *Job Market*: employed and unemployed agents receive new job offers and employed agents are fired, with probabilities depending on the age and SES-specific unemployment rate;
(10) *Relocations*: agents who have accepted job offers from towns different from their current town will relocate with a certain probability;
(11) *Care Transitions*: agents pass to higher care need levels or are hospitalized with a probability depending on age, SES, current care need level and unmet care need.

# 3. Social policy experiments

ABMs allow us to investigate the effects of virtual social or economic policies. This model is characterized by various policy-related parameters, the value of which can be directly or indirectly related to specific measures of social care policy. Therefore, by varying these parameters, we can examine 'what if' scenarios and simulate social care outcomes and costs under alternative social care policies.

For illustrative purposes, we investigated the effect on social care of two policy interventions:

— introduction of fully tax-deductible social care expenses;
— reduction of the government-funded social care eligibility criteria from the Critical care need level to the Substantial care need level (table 1).

In the case of the first policy, the assumption is that it will result in a lower cost of formal social care, as part of the price will be indirectly paid by the government through reduced taxation.[18] Given the progressive taxation system, the price reduction will be higher for the individual earning higher incomes.

The second policy intervention expands the number of people who are eligible to receive government financial support by reducing the required level of care need to receive it (although the means-testing parameters remain the same).

We assume that the two policies are implemented from the simulation year 2020 and compare the outputs of these two policy scenarios with the benchmark no-policy scenario in the period 2020–2040. We present two outputs: the share of unmet care need in each scenario and a measure of the two policies' cost-effectiveness ratio we call *relative incremental cost-effectiveness ratio* (RICER), defined as the ratio between the percentage increase of public expenditure for social care and the percentage increase in the outcome of interest (in our case, the share of the unmet social care need).[19]

---

[18]Income taxation in our model reflects the scheme currently in force in the UK, i.e. tax rates of 40% and 20% for weekly incomes above £663 and £228, respectively (the taxation rate is zero below £228).

[19]Considering that the share of unmet care need is a percentage and that the purpose of social policy is to *decrease* it, we consider the difference between the pre- and the post-intervention shares of unmet social care need. Formally, then, the RICER is computed with the equation:

$$\text{RICER} = \frac{(C_i - C_0/C_0)}{E_0 - E_1},$$

where $C$ is the public expenditure for social care and $E$ is the share of unmet social care need under the policy intervention (i) and the benchmark (0) scenarios. The RICER represents the percentage increase of public expenditure required for 1% percentage decrease of the share of unmet care need.

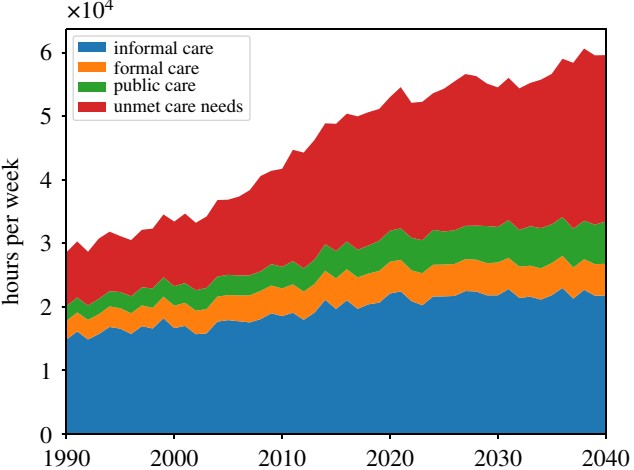

**Figure 1.** Hours of informal and formal care received and of unmet care need in the simulated population.

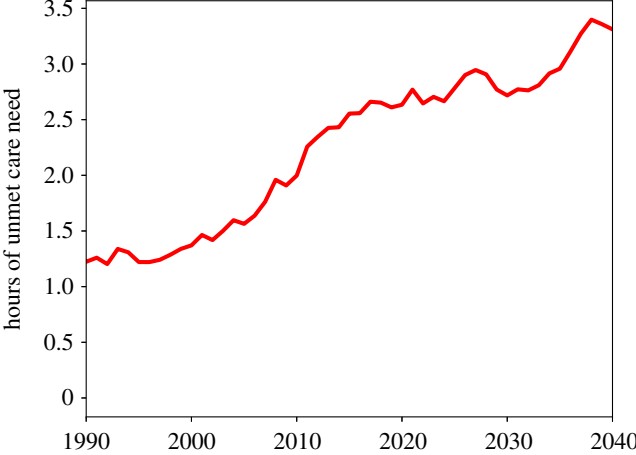

**Figure 2.** Weekly hours of unmet social care need per capita.

# 4. Results

As described above, this simulation is highly complex with numerous processes at play and a number of parameters governing system behaviour. Here we present three sets of results: a series of figures from a representative single run at default parameter values; a series of comparisons of this run, taken as the benchmark, with two simulations where we introduce two alternative policy interventions; and a five-parameter sensitivity analysis. The single-run charts are displayed here in figure 1 through figure 10 and were chosen to highlight the key features of this new simulation framework, namely the modelling of formal and public care,[20] additional economic and labour market details such as socioeconomic status groups, interaction between social care need and healthcare demand, and agent kinship networks.

## 4.1. Benchmark model output

Figure 1 shows the evolution of hours of informal, formal and public care delivered and of unmet social care need per week for the whole population over the period 1990–2040. In our simulations, the total population reaches a peak of around 8100 agents in the period 2027 and then decreases slowly to around 7900 by 2040. As in Bijak *et al.* [22], the simulation's population projections roughly match those of the Office for National Statistics in the UK if we assume no international migration. In figure 1, we can see that after the year 2000, the growth of informal, formal and public care cannot keep

---

[20]To avoid repetition, with the phrase 'formal care', we refer to 'privately paid-for care', whereas with 'public care', we refer to formal care paid for by the local authority.

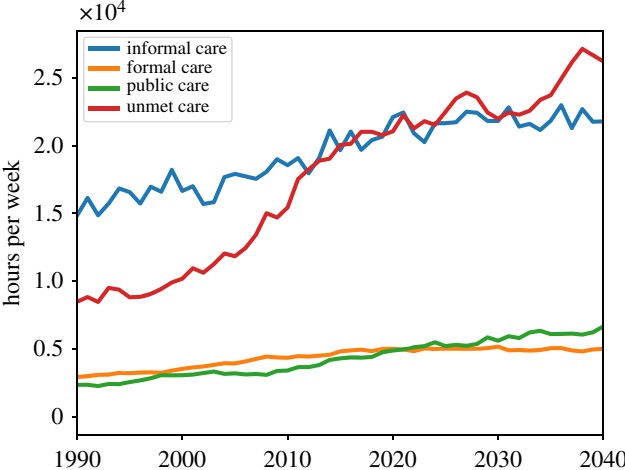

**Figure 3.** Hours of informal, formal and public care received and unmet care need.

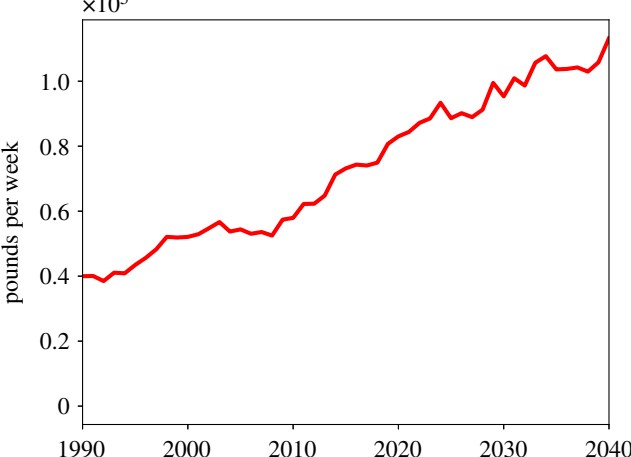

**Figure 4.** Cost of public social care.

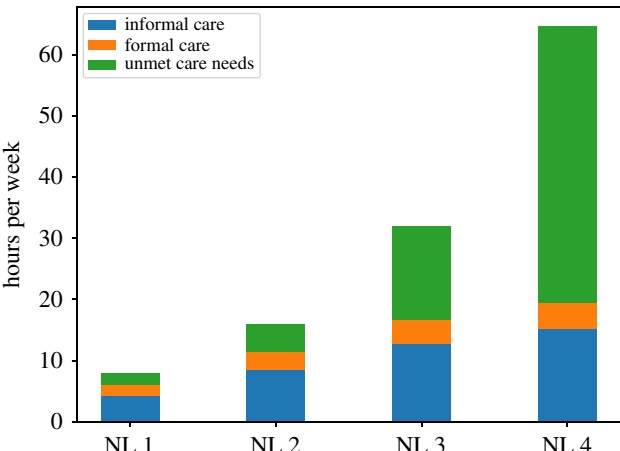

**Figure 5.** Hours of informal and formal care received and unmet care need per recipient by care need level.

pace with the growth in social care need, with a consequent rapid increase of unmet care need, which keeps increasing up to 2040 (with a slight downturn at around 2030). The growth of unmet care need can be clearly seen in figure 2, which shows the dynamics of the *per capita* unmet social care need. Here we can see that the average unmet care need grows from 1990 to 2040 by nearly a factor of 3.

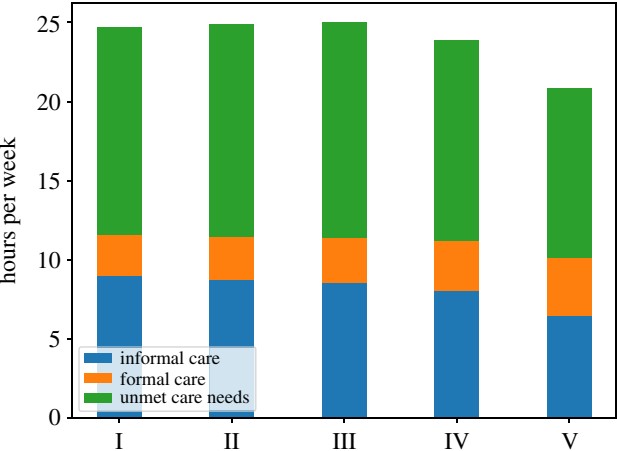

**Figure 6.** Hours of informal and formal care received and unmet care need per recipient by SES group (with I being the poorest SES group).

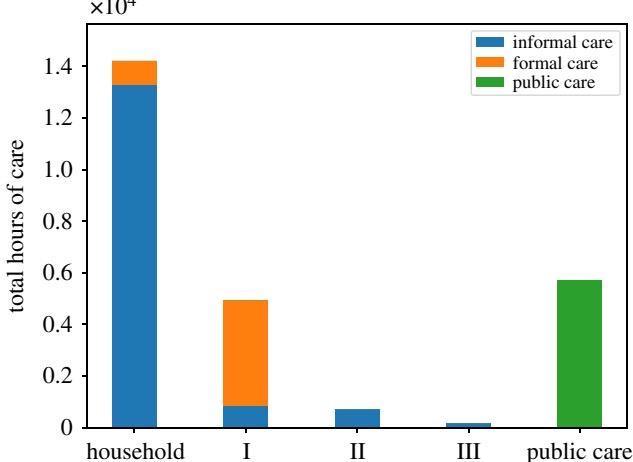

**Figure 7.** Public care and informal and formal care supplied by different groups of receivers' suppliers: household, parents and children (I), grandparents, grandchildren and brothers (II) and aunts, uncles and nephews (III).

Figure 3 shows the dynamics of total informal, formal and public care and of the unmet social care need. We can see that while all three kinds of care increase over time (with the highest growing rate being that of the informal care), their growth is lower than the growth of unmet social care need. Figure 4 shows more of the dynamics of the cost of public social care with the current public care scheme. We can see that, in line with the average unmet care need, it increased by almost a factor of 3 between 1990 and 2040.

Figure 5 shows the mean informal and formal care received and the unmet care need by care need level over the period 2020–2040 (note that the heights of the bars correspond to the weekly hours of care required as shown in table 1). These results show that most of the unmet care need is due to those agents with the highest care need. Figure 6 shows the mean informal and formal care received and the unmet care need by SES group (with I being the poorest SES group). The relative weight of informal care with respect to formal care decreases from the poorest to the wealthiest SES group, a result which reflects empirical findings. As expected, by comparing the green bars in figure 6, we can see that the poorest SES group has a somewhat higher level of unmet care need compared to the wealthiest group but also that unmet care need is a significant share of total care need across all SES groups.

Figure 7 shows the total public care and informal and formal care provided in the period 2020–2040 according to kinship distance: household (distance 0); parents and children (distance 1); grandparents, grandchildren and brothers (distance 2); aunts, uncles and nephews (distance 3). Here we see that most social care is supplied within the household; however, a significant part comes from outside the

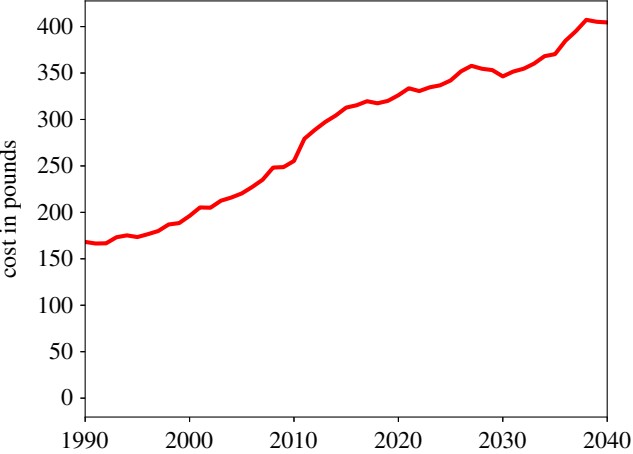

**Figure 8.** *Per capita* healthcare cost due to the hospitalization of people with social care needs (per year).

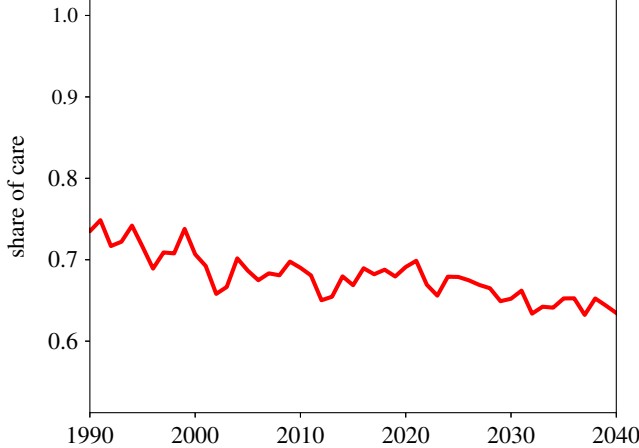

**Figure 9.** Informal care supplied by women as a share of total informal care supplied, at the population level.

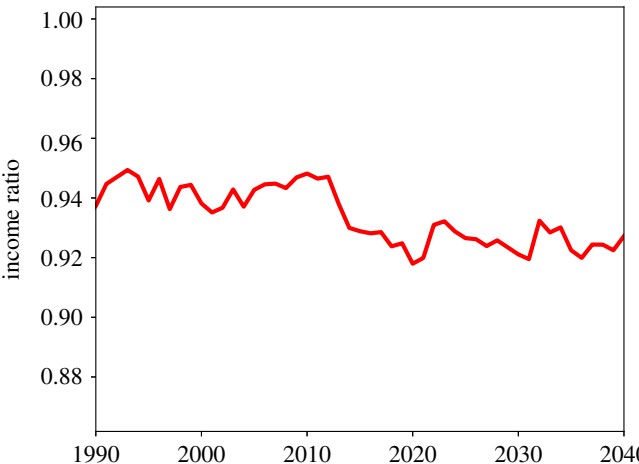

**Figure 10.** Ratio of female to male income at the population level.

household as well, particularly formal care. Figure 8 shows annual *per capita* healthcare cost due to the hospitalization of people with care needs. Healthcare cost grows at a roughly steady rate, following the dynamics of the *per capita* unmet social care need (shown in figure 2). The simulations show that the *per capita* hospitalization cost doubles from the year 2000 to 2040.

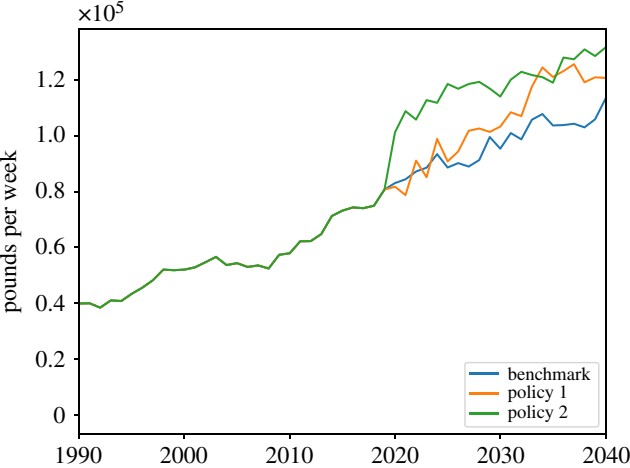

**Figure 11.** Cost of public social care.

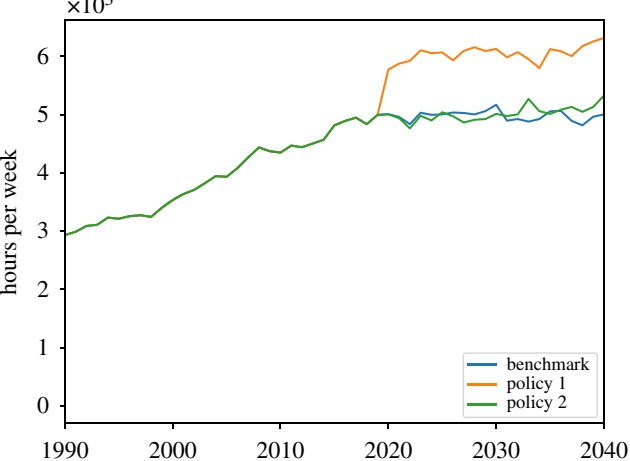

**Figure 12.** Total formal social care.

Figure 9 shows the share of total informal care provided by women. At the population level, this figure starts from just below 75% in 1990, then decreases steadily. Between 2010 and 2020, it fluctuates between 70% and 65%, values which are roughly in line with empirical findings. The decreasing share of informal care provided by women, however, does not produce a positive effect on the gender pay gap, which keeps increasing as shown in figure 10. At the population level, female income is about 6–8% lower than male income. Therefore, our model can explain at least part of the gender pay gap.

## 4.2. Policy intervention output and evaluations

In the next six charts, we investigate and compare the effects and the efficacy of two policy interventions. These two policy experiments are simple, illustrative examples that aim to test how this model may be used as a tool for the design and evaluation of social care policies. As explained in the previous section, we consider a *tax deduction* policy, in which the households are allowed to deduct all the social care expenses from their tax base, and a *direct funding* policy, in which the level of social care needs determining the eligibility for public care is decreased from the *critical* level to the *substantial* level (table 1). The policies are implemented starting in the simulation year 2020, and the two scenarios are compared to the benchmark (i.e. the *no-policy*) scenario.

Figures 11 and 12 show the dynamics of the three scenarios of the cost of public care and the amount of formal social care delivered. In figure 11, we see that relaxation of the public care eligibility criteria leads to an increase in public care cost. Figure 12 shows that the 100% tax deduction of the social care

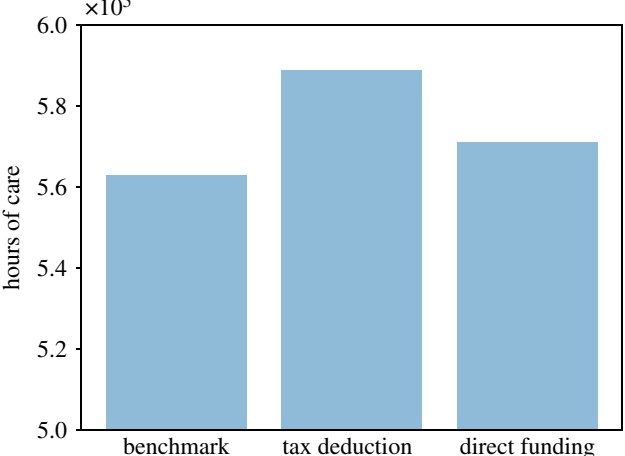

**Figure 13.** Total social care received.

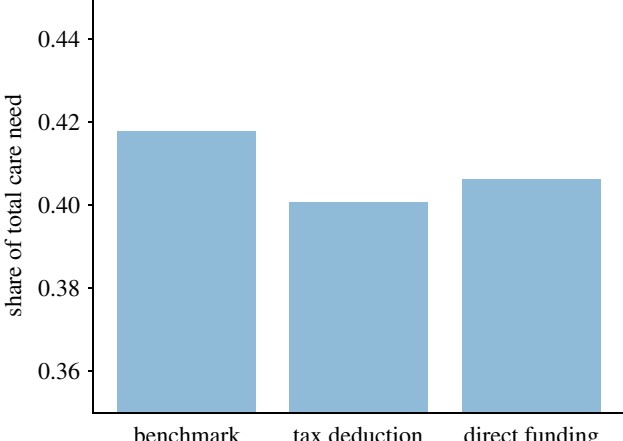

**Figure 14.** Share of unmet social care.

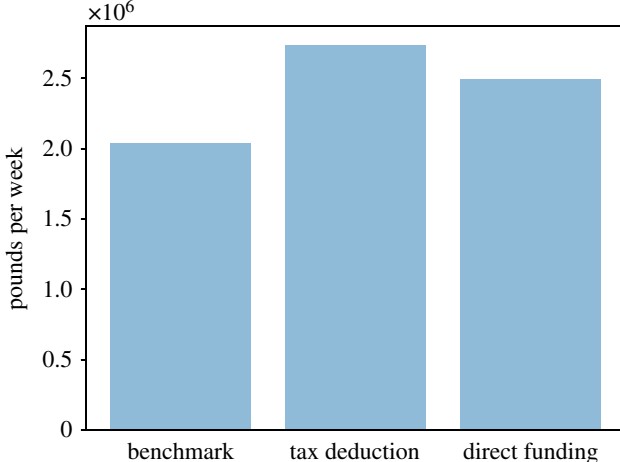

**Figure 15.** Total policy cost for each policy scenario.

expenditures leads to an increase in formal care delivered, as the tax deduction is tantamount to a state contribution to the cost of privately paid-for care, so that it becomes cheaper for families to buy formal care.

In figure 13, we can see that the two policies have a positive impact on total social care received, with the tax deduction intervention having a stronger effect than the relaxation of public care eligibility. This

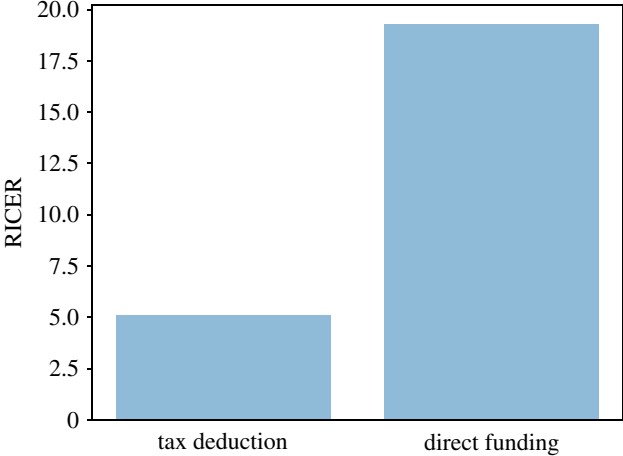

**Figure 16.** Cost-effectiveness of 'tax deduction' and 'direct funding' policy interventions.

finding is confirmed by figure 14, which shows the share of unmet social care need. The benchmark scenario is characterized by the highest share of unmet social care need, while the tax deduction intervention has the lowest share of unmet care need. Overall, we observe that both the tax deduction and the eligibility criteria relaxation policies have a relatively small effect on total unmet social care need, as they reduce the share of unmet care just 1.5% and 1%, respectively, compared to the benchmark.

In order to compare the effectiveness of policy interventions, we must also investigate the cost of their implementation. Figure 15 shows that the tax deduction intervention, although it is the intervention with the greatest effect, is also the most expensive. Therefore, to compare the cost-effectiveness of the two policy interventions we consider their relative ICER (which we refer to as RICER): the policy with the *lowest* RICER will be the most cost-effective one. In figure 16, we observe that the 100% tax deduction intervention is the most cost-effective intervention: with this intervention, we can get a 1% reduction in the share of unmet care need with an increase of about 5% in public expenditure, whereas with the relaxation of the public care eligibility criteria, the same reduction in the share of unmet social care need requires an increase of about 19% of the pre-intervention public expenditure for social care.

## 4.3. Sensitivity analysis with Gaussian process emulators

Here we present the results of a sensitivity analysis conducted with five parameters using a Gaussian process emulator. Gaussian process emulators are, in brief, a means of meta-modelling in which a computer simulation is conceptualized as a function of input parameters to simulation outputs, with those outputs distributed as a multivariate Gaussian distribution [23]. In effect, the emulator produces a statistical model of the initial computational model which can be run orders of magnitude faster than the original, producing in-depth sensitivity and uncertainty analyses with a low computational cost [24].

Meta-modelling methods like Gaussian process emulators save significant time when analysing any simulation that takes more than a few minutes to complete a run, by reducing the number of full simulation runs required. In our case, the simulation can take more than a day to run, meaning that even with these time savings these results took six weeks to calculate using 27 CPU cores on a high-performance workstation.

In using the emulator, we assume that our main simulation output of interest can be decomposed into a series of effects related to our input parameters, as well as interaction effects between those parameters, plus a constant term. In this case, we have also included an additional term which accounts for numerical uncertainty in the computer code itself, sometimes called the 'nugget' term. The emulator was run using the free GEM-SA (Gaussian Emulation Machine for Sensitivity Analysis) software developed by Marc Kennedy (http://www.tonyohagan.co.uk/academic/GEM/).

The five parameters used in this analysis were chosen for their potential importance in the model's dynamics, as each one plays a pivotal role in one of the model's key sub-processes. The five parameters we chose are as follows:

— *unmetCareNeedBias*: regulates the effect of unmet care need on mortality;
— *unmetNeedExponent*: regulates the effect of unmet care need on the transition probabilities between levels of care need;
— *networkSizeParam*: governs the relationship between the size of a household's kinship network in a given town and their likelihood of choosing that town for relocation;

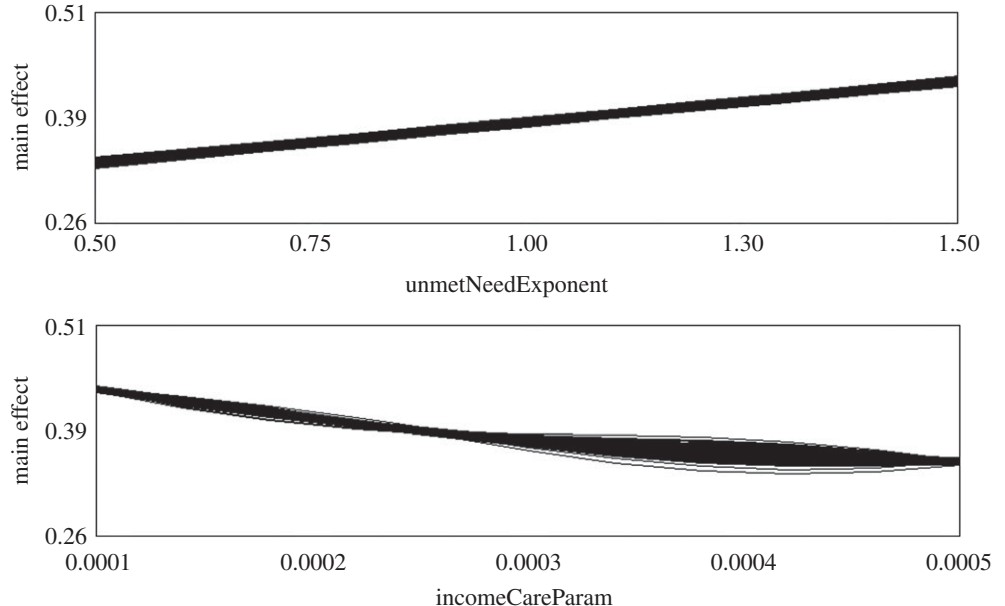

**Figure 17.** Graphs of the main effects of the unmetNeedExponent and incomeCareParam parameters, as produced by the GEM-SA software during sensitivity analysis.

**Table 3.** Values of five chosen parameters and their effect on final output variance.

| parameter name | variance (%) | total effect | high | medium | low |
|---|---|---|---|---|---|
| unmetCareNeedBias | 1.52001 | 1.52034 | 0.7 | 0.5 | 0.3 |
| unmetNeedExponent | 53.5067 | 53.5268 | 1.5 | 1.0 | 0.5 |
| networkSizeParam | 0.259549 | 0.2684 | 15 | 10 | 5 |
| relocationCostParam | 0.0432105 | 0.0434873 | 1.0 | 0.5 | 0.1 |
| incomeCareParam | 44.6411 | 44.6704 | 0.0005 | 0.00025 | 0.0001 |

— *relocationCostParam*: determines the importance of relocation cost in a household's relocation decision (this is parameter $K$ in the relocation cost equation shown in Footnote 11);
— *incomeCareParam*: regulates the effect of the household's *per capita* income on the share of total income that the household allocates to social care provision.

For each parameter, we chose low, medium and high values, then ran one simulation for each possible combination of these values for all five parameters, generating a total of 243 scenarios. For each scenario, we recorded the share of unmet care need in the final year of the simulation as our final output of interest.

Table 3 shows the three values of each parameter chosen for the sensitivity analysis, and the percentage of the final output variance attributed to each parameter by the emulator. The emulator results show that *unmetNeedExponent* and *incomeCareParam* had by far the most effect on the final output of the simulation runs, accounting for 53.51% and 44.64% of the output variance respectively. The emulator was run using the additional 'nugget' term mentioned above, which allows the emulator to account for numerical uncertainty in the computer code. Prior to the inclusion of this term, the total output variance accounted for by these five parameters was just above 80%; once the nugget was included, however, the total climbed to 99.97%. This demonstrates the importance of accounting for numerical code uncertainty (such as rounding errors) in complex simulations.

Figure 17 shows two graphs produced by the GEM-SA software used to perform the sensitivity analysis (axis labels and numerals in these graphs have been modified to enhance readability). These graphs show the main effects of the *unmetNeedExponent* and *incomeCareParam* parameters on the proportion of unmet care need in the agent population in the final year of the simulation. The emulator suggests a positive correlation between the values of *unmetNeedExponent* and the share of unmet care need in the simulated population, and a negative correlation between *incomeCareParam* and the share of unmet need. These

sensitivity analysis results suggest that a greater likelihood of unmet care need leading to transitions into more severe levels of need increases the proportion of unmet need in the simulated population. Conversely, increases in the shares of income spent on social care by households decrease the level of unmet care need.

As with the other results reported here, this analysis is included for illustrative purposes. As the framework develops further and simulation parameters are calibrated using empirical data on social care provision, we can use these methods to examine which simulation parameters are most closely linked to our outputs of interest, and in so doing determine where in the system potential interventions may be most effective.

## 5. Discussion

While the results presented here are still early, the outcomes of these simulation runs suggest that this model can produce broadly realistic portraits of the coming trends in UK social care. The overall population dynamics largely mirror those in evidence in the real world, with the notable exception of international migration which is not modelled here. The simulation also produces inequalities in care provision by both SES and gender; while the data presently available is not sufficient to determine if the socioeconomic inequalities are accurate, the gender pay gap is broadly reflective of current UK norms.

The simulation results on unmet care need lend credence to the worries expressed by UK policy-makers. The simulation shows that unmet care need will continue to grow over time, and given that 1.2 million older people in England alone are not receiving sufficient care [2], further growth in these figures could lead to severe consequences for public health. The results also show marked inequalities in care provision, with the wealthy capable of supporting their aged relatives in need through privately funded care while still staying in work and receiving high wages. Among the lower-income groups, women are providing an overwhelmingly larger share of informal care as compared to men, meaning that women at the lower end of the socioeconomic scale are more likely to be pushed out of work and toward unpaid care provision.

Our use of kinship networks as the mechanism for distributing care illustrates that kinship distance impacts care provision behaviour, and thus is an important aspect of care decision-making that should be taken into account in future research. We observed a significant difference in caring behaviours between within-household and kinship distance I agents, with the latter providing much more formal care than informal. This suggests that future models in this area should consider kinship networks and their impact on caring behaviours. Understanding the negotiation process within families regarding care provision will be an important aspect for policy-makers to examine as well, as policies put into place to support and encourage informal care may need to take account of these complex social aspects of care.

Finally, the speculative policy scenarios we provide here demonstrate the efficacy of this platform as an aid to policy-makers who wish to examine the impact and possible unintended side-effects (spillover effects) of their planned policy interventions. The simulations showed that while some policies such as tax deductions may seem like an easy 'win' for the policy-maker, the actual impact on social care need of even a 100% tax deduction is minimal. At the same time, the relaxation of health-related eligibility criteria is not effective in decreasing unmet need substantially. In order to generate significant reductions in unmet social care need, more expansive—and expensive—interventions are needed, one possible candidate being the relaxation of the economic eligibility criteria. From a methodological point of view, the two policy interventions presented here showed that the model is capable of simulating the results of nuanced policy interventions, due to the detailed modelling of key social care mechanisms and related economic and social processes.

## 6. Future work

While the current simulation does broadly reflect the expected trends in social care provision, validating the results is difficult. Ongoing projects in Scotland are linking administrative data sources to develop a clearer picture of social care provision and receipt across the country. In future revisions of the model, we intend to make use of these data by focusing the simulation on Scotland rather than the whole UK.

Apart from the validation process, this model can be expanded in various directions. First, according to the Office for National Statistics, the proportion of informal childcare to GDP increased from 13.8% to 17.6% in the period 2005–2014 [25]. Child care represents a significant part of the total household's informal care need and thus affects available social care supply. Our next planned update to the model will include a child care mechanism which will allow us to model the interaction with social care supply.

Second, according to the Office for National Statistics, in 2016, 28.2% of births in England and Wales were to women who were not born in the UK [26]. Moreover, projections show that post-2016 immigration accounts for 77% of total population growth until 2041 [27]. The addition of international migration to the model will allow us to understand UK demographic dynamics in future decades, and to more closely replicate UK population change. This will also enable us to explore whether migration policies can help to reduce the 'care gap' produced by dropping birth rates in the UK-born population.

Following on from the simplistic policy comparison presented here, in future work, we will consult with social care policy-makers in Scotland to model proposed social care policies in more detail. Social care policy is very complex, with the programmes available varying not only by region (England, Wales, Scotland and Northern Ireland), but by individual local councils. The model is capable of representing these details by varying relevant policy parameters across the simulated UK geography. Once we are able to replicate the current state of social care policy in the UK, we will then be able to develop detailed evaluations of the possible impacts of new policy interventions. Our model's ability to capture detailed individual-level care decision-making will enable policy-makers to examine policies aimed at behavioural change as well as broader economic and social policies.

Finally, while the current simulation is very UK-centric, the core simulation engine can be altered easily to examine the situation in other countries. Ultimately we hope to produce a simulation framework that is capable of modelling informal and formal social care across a variety of cultural and economic contexts.

Data accessibility. The Python 2.7 code is available on GitHub: https://github.com/UmbertoGostoli/ABM-for-social-care/tree/v.09 [18]. The output data used for the figures are available in this Dryad repository: https://doi.org/10.5061/dryad.h69t34v [28].

Authors' contributions. U.G. developed the model (based on a previous version, developed by E.S.), ran the simulations, gathered the data and produced the charts. E.S. conceived the research, helped develop the model and performed the sensitivity analysis. Both authors contributed to the writing of the manuscript. Both authors gave final approval for publication.

Competing interests. We declare we have no competing interests.

Funding. The authors are part of the Complexity in Health Improvement Programme supported by the Medical Research Council (MC_UU_12017/14) and the Chief Scientist Office (SPHSU14).

Acknowledgements. We would like to thank Chris Patterson and Lauren White from the MRC/CSO Social and Public Health Sciences Unit for their helpful comments.

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
