## [Reviewer comments · Royal Society Open Science]

Review History

RSOS-190029.R0 (Original submission)

Review form: Reviewer 1

Is the manuscript scientifically sound in its present form?

Yes

Are the interpretations and conclusions justified by the results?

Yes

Is the language acceptable?

Yes

Is it clear how to access all supporting data?

Yes

Do you have any ethical concerns with this paper?

No

Have you any concerns about statistical analyses in this paper?

No

Recommendation?

Accept with minor revision (please list in comments)

Comments to the Author(s)

Please see attached note (Appendix A).

Review form: Reviewer 2 (Pietro Terna)

Is the manuscript scientifically sound in its present form?

Yes

Are the interpretations and conclusions justified by the results?

Yes

Is the language acceptable?

Yes

Is it clear how to access all supporting data?

Yes

Do you have any ethical concerns with this paper?

No

Have you any concerns about statistical analyses in this paper?

No

Recommendation?

Accept with minor revision (please list in comments)

Comments to the Author(s)

The paper is so important and significant that it is worth minor integrations and of future relevant improvements.

Minor integrations: (i) add a simple scheme of the simulation, showing the sequence of the action within each "year" and the agents involved; (ii) specify that the code is related to Python2; (iii) create a list of required libraries to run the code (a requirements.txt file within the GitHub would be sufficient).

Future development (to be declared as integration to the current version): (a) have the capability of experimenting with different birth rates, which is very important for the future; (b) the same, for migration rates; (c) the same, with working hours, considering the probability of facing aging with different production structures (e.g., more automation); (d) introduce imitation among agents as a way to modify their behavior, with spreading effects.

Without (d) the whole model is closer to a microsimulation than to an agent-based simulation.

Review form: Reviewer 3

Is the manuscript scientifically sound in its present form?

Yes

Are the interpretations and conclusions justified by the results?

No

Is the language acceptable?

Yes

Is it clear how to access all supporting data?

Yes

Do you have any ethical concerns with this paper?

No

Have you any concerns about statistical analyses in this paper?

No

Recommendation?

Major revision is needed (please make suggestions in comments)

Comments to the Author(s)

The paper extends an earlier work of Silverman et al. [14] by providing an augmented agent-based model of social care benchmarked to the UK population dynamics, and by analysing some of the resulting policy implications in two alternative scenarios. The key innovations of the paper are threefold: (1) including the socio-economic status of agents, (2) embedding the kinship networks and "social distance" between agents, and (3) introducing elements of the formal care market and labour market. All of these innovations are non-trivial, make the model more realistic and better suited to addressing actual policy questions, and in my view warrant a separate publication. Replicability of the results is ensured via two online repositories, one with the model code, the other one with the data.

Where the paper falls short, even taking a step back towards the predecessor [14], is the analytical layer. The conclusions mention in passing the need for sensitivity analysis, but in order to make the findings robust, this should have been a part of the current analysis, alongside with the basic uncertainty analysis. Of particular interest would be the relative impacts of a set of free parameters (such as F, I, K, R, and other) on the key outputs of interest (ICER, unmet care need, informal care). In short, a formal analysis of systematic variation of outcomes across the free parameter space is badly needed, also from a policy point of view.

The description of the model also needs a more detailed discussion of the two policy scenarios (tax deduction and direct funding), including any policy-specific parameters (tax rates?), which may also be subject to sensitivity testing. If space is constrained, then some space could be freed up by condensing the description of the pre-existing model [14] in Section 2. From the current description, it is not clear how the money transfers work in the system - and of course the tax rate will affect the demand for formal care by changing disposable income.

The model, and the discussion of policy implications of the model also misses an elephant in the room, which is the international migration component. While I do not think that adding it is absolutely necessary, what the results seem to be pointing at is a care gap, some of which can be met through migration (which in turn is a policy, as well as a political choice). One additional

analysis that could help shed light on that might be along the lines of "replacement migration" - how many migrants from outside the system would be needed in the care sector to fill the gaps, under some realistic assumptions regarding migrant wages relative to the local labour force.

Finally, a minor, stylistic point: the introduction starts with a sombre tone, but it is actually worth mentioning that the care challenges are actually a byproduct of one of the greatest successes of humanity so far, which is the increase in longevity (and also in years spent in good health).

Decision letter (RSOS-190029.R0)

08-Apr-2019

Dear Dr Silverman,

The editors assigned to your paper ("Modelling Social Care Provision in an Agent-Based Framework with Kinship Networks") have now received comments from reviewers. We would like you to revise your paper in accordance with the referee and Associate Editor suggestions which can be found below (not including confidential reports to the Editor). Please note this decision does not guarantee eventual acceptance.

Please submit a copy of your revised paper before 01-May-2019. Please note that the revision deadline will expire at 00.00am on this date. If we do not hear from you within this time then it will be assumed that the paper has been withdrawn. In exceptional circumstances, extensions may be possible if agreed with the Editorial Office in advance. We do not allow multiple rounds of revision so we urge you to make every effort to fully address all of the comments at this stage. If deemed necessary by the Editors, your manuscript will be sent back to one or more of the original reviewers for assessment. If the original reviewers are not available, we may invite new reviewers.

- Data accessibility

If you wish to submit your supporting data or code to Dryad (<http://datadryad.org/>), or modify your current submission to dryad, please use the following link:
<http://datadryad.org/submit?journalID=RSOS&manu=RSOS-190029>

- Competing interests

- Authors' contributions

- Acknowledgements

- Funding statement

on behalf of Dr Hamed Haddadi (Associate Editor) and Professor Marta Kwiatkowska (Subject Editor)

Comments to Author:

Reviewers' Comments to Author:

Reviewer: 1

Comments to the Author(s)

Please see attached note

Reviewer: 2

Comments to the Author(s)

The paper is so important and significant that it is worth minor integrations and of future relevant improvements.

Minor integrations: (i) add a simple scheme of the simulation, showing the sequence of the action within each "year" and the agents involved; (ii) specify that the code is related to Python2; (iii) create a list of required libraries to run the code (a requirements.txt file within the GitHub would be sufficient).

Future development (to be declared as integration to the current version): (a) have the capability of experimenting with different birth rates, which is very important for the future; (b) the same, for migration rates; (c) the same, with working hours, considering the probability of facing aging with different production structures (e.g., more automation); (d) introduce imitation among agents as a way to modify their behavior, with spreading effects.

Without (d) the whole model is closer to a microsimulation than to an agent-based simulation.

Reviewer: 3

Comments to the Author(s)

The paper extends an earlier work of Silverman et al. [14] by providing an augmented agent-based model of social care benchmarked to the UK population dynamics, and by analysing some of the resulting policy implications in two alternative scenarios. The key innovations of the paper are threefold: (1) including the socio-economic status of agents, (2) embedding the kinship networks and "social distance" between agents, and (3) introducing elements of the formal care market and labour market. All of these innovations are non-trivial, make the model more realistic and better suited to addressing actual policy questions, and in my view warrant a separate publication. Replicability of the results is ensured via two online repositories, one with the model code, the other one with the data.

Where the paper falls short, even taking a step back towards the predecessor [14], is the analytical layer. The conclusions mention in passing the need for sensitivity analysis, but in order to make the findings robust, this should have been a part of the current analysis, alongside with the basic uncertainty analysis. Of particular interest would be the relative impacts of a set of free parameters (such as F, I, K, R, and other) on the key outputs of interest (ICER, unmet care need, informal care). In short, a formal analysis of systematic variation of outcomes across the free parameter space is badly needed, also from a policy point of view.

The description of the model also needs a more detailed discussion of the two policy scenarios (tax deduction and direct funding), including any policy-specific parameters (tax rates?), which

may also be subject to sensitivity testing. If space is constrained, then some space could be freed up by condensing the description of the pre-existing model [14] in Section 2. From the current description, it is not clear how the money transfers work in the system - and of course the tax rate will affect the demand for formal care by changing disposable income.

The model, and the discussion of policy implications of the model also misses an elephant in the room, which is the international migration component. While I do not think that adding it is absolutely necessary, what the results seem to be pointing at is a care gap, some of which can be met through migration (which in turn is a policy, as well as a political choice). One additional analysis that could help shed light on that might be along the lines of "replacement migration" - how many migrants from outside the system would be needed in the care sector to fill the gaps, under some realistic assumptions regarding migrant wages relative to the local labour force.

Finally, a minor, stylistic point: the introduction starts with a sombre tone, but it is actually worth mentioning that the care challenges are actually a byproduct of one of the greatest successes of humanity so far, which is the increase in longevity (and also in years spent in good health).

Author's Response to Decision Letter for (RSOS-190029.R0)

See Appendix B.

Decision letter (RSOS-190029.R1)

28-May-2019

Dear Dr Silverman:

On behalf of the Editors, I am pleased to inform you that your Manuscript RSOS-190029.R1 entitled "Modelling Social Care Provision in an Agent-Based Framework with Kinship Networks" has been accepted for publication in Royal Society Open Science subject to minor revision in accordance with the referee suggestions. Please find the comments at the end of this email.

The Associate Editor and Subject Editor have recommended publication, but also suggest some minor revisions to your manuscript. Therefore, I invite you to respond to the comments and revise your manuscript.

- Ethics statement

- Data accessibility

It is a condition of publication that all supporting data are made available either as supplementary information or preferably in a suitable permanent repository. The data accessibility section should state where the article's supporting data can be accessed. This section should also include details, where possible of where to access other relevant research materials such as statistical tools, protocols, software etc can be accessed. If the data has been deposited in

an external repository this section should list the database, accession number and link to the DOI for all data from the article that has been made publicly available. Data sets that have been deposited in an external repository and have a DOI should also be appropriately cited in the manuscript and included in the reference list.

<http://datadryad.org/submit?journalID=RSOS&manu=RSOS-190029.R1>

- **Competing interests**

- **Authors' contributions**

- **Acknowledgements**

- **Funding statement**

Because the schedule for publication is very tight, it is a condition of publication that you submit the revised version of your manuscript before 06-Jun-2019. Please note that the revision deadline will expire at 00.00am on this date. If you do not think you will be able to meet this date please let me know immediately.

on behalf of Dr Hamed Haddadi (Associate Editor) and Marta Kwiatkowska (Subject Editor)
openscience@royalsociety.org

Associate Editor Comments to Author (Dr Hamed Haddadi):

Dear authors,
I am happy with the improvements made in addressing the reviewer comments. There are small improvements that can still be made in quality of figures (e.g., legible axes labels for figures 11-12, better version of figure 17) and the explanations in the paper, after which the paper can be accepted for publications. Please perform these and prepare a final version of the paper.

Best wishes

Author's Response to Decision Letter for (RSOS-190029.R1)

See Appendix C.

Decision letter (RSOS-190029.R2)

21-Jun-2019

Dear Dr Silverman,

I am pleased to inform you that your manuscript entitled "Modelling Social Care Provision in an Agent-Based Framework with Kinship Networks" is now accepted for publication in Royal Society Open Science.

Kind regards,

on behalf of Dr Hamed Haddadi (Associate Editor) and Marta Kwiatkowska (Subject Editor)
openscience@royalsociety.org

Associate Editor Comments to Author (Dr Hamed Haddadi):

The reviewers' comments have been addressed and the article is ready to proceed.

Appendix A

Comments on Gostoli and Silverman paper on Modelling Social Care Provision in an Agent-Based Framework with Kinship Networks

This paper presents an interesting simulation of care provision in the U.K. in which family members decide to provide unpaid care or purchase private care for their relative who has care needs. It is an interesting and valuable paper on a policy-relevant and important topic.

General points:

It is surprising that formal care is not discussed or defined and that local authority funded care is not mentioned until page 8. It appears that formal care is treated in this paper solely as private care bought by relatives. Since much home care in the UK is funded by local authorities the role of publicly funded care should be discussed.

The importance of attitudes and preferences in the choice of types of care sought and provided should be mentioned even though they cannot be modelled.

Introduction:

The projections in Wittenberg and Hu (ref 6) are on a set of base case assumptions: this should be stated.

The Health Survey for England (HSE) provides data on receipt of unpaid care by older people and on provision of unpaid care by adults of any age.

It is important to distinguish between care which is fully privately funded and care where the family are providing 'top up' payments to local authority support.

The model: health status and care need

Those with critical needs are likely to need to be in care homes, which are not mentioned.

The model: model enhancements: kinship networks

The number of hours of care which agents can provide (table 2) seems rather high in some of the categories. Studies have shown that unpaid care of more than 10 hours per week, and certainly more than 20 hours per week, is very difficult to combine with full-time work. The hours for teenagers and students also seem high. Account should also be taken of potential carers' other caring responsibilities, especially caring for their children.

It is not clear what role estimated informal care attraction plays in the model given the statement (footnote 9) that relocation is prompted only by a job offer or marriage.

The model: model enhancements: formal care:

My comments above about formal care and its definition are relevant here. The local authority may fund the formal care or the person with care needs could do so.

Social care experiments:

It is not clear whether social care expenses of £X means that £X is deducted from taxable income or £x is deducted from the tax bill – presumably the former but the drafting suggests the latter. There is of course direct public funding of care in the UK subject to eligibility criteria and (except in Scotland) a means test.

Results

It is not initially clear that figures 1-8 relate to the base case without any policy experiment.

It is surprising that total population peaks in 2025 which does not seem consistent with ONS population projections. This requires an explanation.

The final part of the results section, on the impact of the two policy experiments, is somewhat brief. It is not clear, for example, how the cost to public funds of the two policy experiments is estimated how the ICERs are estimated, or whether the ISERs are estimated using a public sector or societal perspective.

Discussion

Some data on inequalities by SEG in receipt of home care by older people is available from the HSE, and there have been some studies on this topic.

The finding that 'kinship distance impacts care provision behaviour' is more an underlying assumption of the modelling than a finding.

The limitations of the modelling, and its strengths, would normally be discussed in this section.

Future work

The outline of plans for future work is a valuable part of the paper. Scotland is however rather different from England, Wales and N Ireland in terms of the generosity of its funding system (free personal care for older people). But the greater availability of data for Scotland means that the plan to start with Scotland seems sensible.

Appendix B

Reviewer Response: Modelling Social Care Provision in An Agent-Based Framework with Kinship Networks

Umberto Gostoli and Eric Silverman

1 Summary

We have attempted to address every comment from the three reviewers, and below we describe how we have done so. The most significant changes are the addition of publicly-funded care to the model, which required re-doing all the default and policy comparison simulation runs, and the addition of a sensitivity analysis using Gaussian process emulators, which required weeks of computation. This revised version of the paper now includes a more nuanced model of care provision, more detailed analyses of the outcomes, and greater clarity in our exposition and interpretation. We would like to thank all three reviewers for their detailed comments, which have substantially improved the paper.

2 Reviewer One Comments

It is surprising that formal care is not discussed or defined and that local authority funded care is not mentioned until page 8. It appears that formal care is treated in this paper solely as private care bought by relatives. Since much home care in the UK is funded by local authorities the role of publicly funded care should be discussed.

We have moved the definition of social care further up in the paper. We have also enhanced the current version of the model by including local-authority-funded care, and have redone all the results accordingly. We have described the implementation of this in the Methods section. We had not initially planned to add this at this stage, but doing so turned out to be less difficult than we had feared, so we decided to add this to the current paper in order to address this criticism.

The importance of attitudes and preferences in the choice of types of care sought and provided should be mentioned even though they cannot be modelled.

We have mentioned both in the *Motivations* subsection and in the *Future Work* section that attitudes and preferences are another aspect of behavioural modelling that may be introduced in later versions of this model. We disagree that they cannot be modelled, though certainly it would be challenging.

The projections in Wittenberg and Hu (ref 6) are on a set of base case assumptions: this should be stated. The Health Survey for England (HSE) provides data on receipt of unpaid care by older people and on provision of unpaid care by adults of any age.

We have added in a footnote the requested statement about the assumptions on which the projections in Wittenberg and Hu (2015) are based.

We have also added the Health Survey for England 2017 reference, some findings of which we added in the *Introduction* section, and thank the reviewer for the suggestion.

It is important to distinguish between care which is fully privately funded and care where the family are providing top up payments to local authority support.

The new version of the model includes both varieties of formal care.

Those with critical needs are likely to need to be in care homes, which are not mentioned.

We have now mentioned this aspect in a footnote in *Health Status and Care Need*: “Note that in the UK, people with critical care needs are likely to be placed into care homes. In this model we do not explicitly represent this aspect, but, for the sake of simplicity, the social care received in a care home is implicitly considered as part of the formal social care received by people in need. Future iterations of the model will simulate care homes more explicitly.”

The number of hours of care which agents can provide (table 2) seems rather high in some of the categories. Studies have shown that unpaid care of more than 10 hours per week, and certainly more than 20 hours per week, is very difficult to combine with full-time work. The hours for teenagers and students also seem high. Account should also be taken of potential carers other caring responsibilities, especially caring for their children. It is not clear what role estimated informal care attraction plays in the model given the statement (footnote 9) that relocation is prompted only by a job offer or marriage.

Our perusal of relevant studies in Scotland suggest these figures are not out of the realm of possibility. However, following the reviewer’s suggestion, we have significantly reduced the hours of care of students and employed agents in this updated version of the model. We would stress again that this model, as explicitly described in the *Motivations* section, is not intended to be highly accurate or predictive, but illustrative of what this framework can simulate and the behaviours it generates. We are in contact with social care experts here in Scotland who will be helping us to refine these aspects of the simulation.

It is not clear whether social care expenses of X means that X is deducted from taxable income or x is deducted from the tax bill presumably the former but the drafting suggests the latter. There is of course direct public funding of care in the UK subject to eligibility criteria and (except in Scotland) a means test.

This point has now been clarified.

It is not initially clear that figures 1-8 relate to the base case without any policy experiment.

We have made this more explicit in the text by altering the last sentence of the first paragraph of *Results*: “The single-run charts are displayed here in Figures 1 through 8, and were chosen to highlight the key features of this new simulation framework...”

We have also divided the Results into subsections, to make it more clear where we are examining the base case and the policy comparisons.

It is surprising that total population peaks in 2025 which does not seem consistent with ONS population projections. This requires an explanation.

As noted in the text, the model does not include international migration, and as such we do not intend, nor do we claim to precisely replicate UK population change. As noted throughout we aimed for this stage of the model development to provide qualitative similarity to real-world behaviour, and have no pretensions toward predictive accuracy at this time. Developing a model of this complexity to a high level of predictive accuracy is a very long-term project.

We have underlined this point further with the following sentence in *Future Work*: “The addition of international migration to the model will allow us to understand UK demographic dynamics in future decades, and to more closely replicate UK population change.”

The final part of the results section, on the impact of the two policy experiments, is somewhat brief. It is not clear, for example, how the cost to public funds of the two policy experiments is estimated how the ICERs are estimated, or whether the ISERs are estimated using a public sector or societal perspective.

Details on the ICER calculations are now included in the Results section.

Some data on inequalities by SEG in receipt of home care by older people is available from the HSE, and there have been some studies on this topic. The finding that kinship distance impacts care provision behaviour is more an underlying assumption of the modelling than a finding. The limitations of the modelling, and its strengths, would normally be discussed in this section.

Regarding the kinship distance impact, we present this in the Discussion as underlining the importance of modelling this aspect, in that the model’s results

show the effects are non-trivial. Our initial assumptions do drive some of this, namely the restriction on informal care provision linked to kinship distance, but the simulation outcomes are also driven by emergent effects from the interactions between kinship distance, spatial distance, and related agent behaviours.

We have worked to be very clear throughout the paper about the motivations of the project and the limitations of the current model, as well as what additional factors are going to be examined in future iterations. We have not added additional discussion on this issue, as we felt that further repetition of the model's caveats would bog the paper down rather than provide any additional clarity.

3 Reviewer Two Comments

****The paper is so important and significant that it is worth minor integrations and of future relevant improvements.**

We thank the reviewer for this encouraging comment.

Minor integrations: (i) add a simple scheme of the simulation, showing the sequence of the action within each "year" and the agents involved; (ii) specify that the code is related to Python2; (iii) create a list of required libraries to run the code (a requirements.txt file within the GitHub would be sufficient).

We agree that these additions would be helpful, and have added a summary of the sequence of each yearly update in the simulation (Section 2, subsection e, sub-subsection vii).

Future development (to be declared as integration to the current version): (a) have the capability of experimenting with different birth rates, which is very important for the future; (b) the same, for migration rates; (c) the same, with working hours, considering the probability of facing aging with different production structures (e.g., more automation); (d) introduce imitation among agents as a way to modify their behavior, with spreading effects.

All of the mentioned aspects are user-alterable, as they are driven by input data and parameters. However in most cases we expect most of these to be left unchanged by users, as they will want to use empirical values for birth rates, migration, etc. as starting points.

In the *Future Work* section, we have added mentions of additional 'what-if' scenarios, including automation, and the possibility of further behavioural influences between agents. The automation idea is beyond the scope of our current projects, but we hope mentioning it here may inspire others to take this forward.

4 Reviewer Three Comments

Where the paper falls short, even taking a step back towards the predecessor [14], is the analytical layer. The conclusions mention in passing the need for sensitivity analysis, but in order to make the findings robust, this should have been a part of the current analysis, alongside with the basic uncertainty analysis. Of particular interest would be the relative impacts of a set of free parameters (such as F, I, K, R, and other) on the key outputs of interest (ICER, unmet care need, informal care). In short, a formal analysis of systematic variation of outcomes across the free parameter space is badly needed, also from a policy point of view.

As in the preceding paper cited by the reviewer, we have now utilised Gaussian process emulation to perform a sensitivity analysis using the five parameters identified in this comment. Embarking on a sensitivity analysis for a simulation of this complexity is a major undertaking, as an individual run takes 24-48 hours to complete. Producing enough simulation output data to run the emulator required 243 simulation runs, which took six weeks to complete on our high-performance workstation.

Note that we have only used unmet care need as an output of interest, as running a emulator capable of coping with a vector of outputs would vastly increase the amount of runs needed and the complexity of the procedure, and would have been impossible to perform in the time allotted for revisions, and indeed probably impossible in general until our computing resources are significantly increased. Very few modellers have attempted Gaussian process emulators of this type, so we also would have needed to write new software for this purpose. We will take a look at implementing analyses of this complexity when we are developing future work intended to provide more concrete policy recommendations, rather than illustrative comparisons of simplified exemplar policies as in this paper.

We have added a new subsection to the Results section, subsection c, which introduces Gaussian process emulation, describes our sensitivity analysis in detail, presents a table and two graphs of results, and provides our interpretation of the outcomes.

The description of the model also needs a more detailed discussion of the two policy scenarios (tax deduction and direct funding), including any policy-specific parameters (tax rates?), which may also be subject to sensitivity testing.

We have added some additional details on the two policy scenarios presented. We have refrained from a detailed sensitivity analysis in this case, as these scenarios are intended as illustrative of model behaviour and outputs and not as a serious attempt at policy evaluation (and we have highlighted this fact in this revised manuscript). More complex policy scenarios will be examined in detail with sensitivity analyses in future work.

The model, and the discussion of policy implications of the model also misses an elephant in the room, which is the international migration component.

We agree that this is a very useful direction for future work, and have referred to the care gap and international migration in the *Future Work* section in this sentence: “Moreover, this will enable us to explore whether migration policies can help to reduce the ‘care gap’ produced by dropping birthrates in the UK-born population.”

Finally, a minor, stylistic point: the introduction starts with a sombre tone, but it is actually worth mentioning that the care challenges are actually a byproduct of one of the greatest successes of humanity so far, which is the increase in longevity (and also in years spent in good health).

We agree with this stylistic suggestion and have added a new first sentence to the *Introduction* which reads as follows: “In recent history, researchers and practitioners in public health have succeeded in significantly lengthening the human lifespan and increasing quality of life for the elderly.”

Appendix C

Dr Umberto Gostoli and Dr Eric Silverman
MRC/CSO Social and Public Health Sciences Unit
University of Glasgow
200 Renfield Street
Glasgow G2 3AX

Dear Dr Haddadi,

Please find attached our final revised manuscript for the paper entitled Modelling Social Care Provision in an Agent-Based Framework with Kinship Networks. As requested, we have modified Figure 17 for enhanced readability and clarity, we have revisited our textual explanations of all 17 figures in the text, and we have modified all axes labels in all of the figures to ensure legibility.

We hope you find these modifications to your satisfaction and look forward to appearing in your pages.

Yours sincerely,

Eric Silverman and Umberto Gostoli